# Self-allocation bias in performance-based cooperative decisions is driven by self-interest rather than distorted performance encoding

Sihui Zhang[1,2], Xue Yong[1], Yina Ma[1,3‡]*, Christoph W. Korn[2,4‡]*

**1** State Key Laboratory of Cognitive Neuroscience and Learning & IDG/McGovern Institute for Brain Research, Faculty of Psychology, Beijing Normal University, Beijing, China, **2** Section Social Neuroscience, Department of General Psychiatry, University of Heidelberg, Heidelberg, Germany, **3** Chinese Institute for Brain Research, Beijing, China, **4** Institute for Systems Neuroscience, University Medical Center Hamburg-Eppendorf, Hamburg, Germany

‡ These authors share last authorship on this work.
* yma@bnu.edu.cn (YM); christoph.korn@med.uni-heidelberg.de (CWK)

## Abstract

Human cooperation often involves performing joint tasks, where success relies on how collective rewards are allocated among cooperating parties based on their individual performance and contribution to task outcomes. However, it remains unclear whether and how individual performance and contribution give rise to self-related biases in such allocation decisions. Here, we developed a novel performance-based social allocation task that manipulated how individual performance contributed to joint outcomes. Across two experiments, participants exhibited a robust self-allocation bias: they allocated more rewards to themselves and disproportionately disregarded their own performance, particularly when their performance did not causally contribute to the joint outcome. This self-allocation bias was amplified in individuals with stronger individualistic social preferences, as measured by social value orientation. At the neural level, self-relevant (versus self-irrelevant) allocation decisions were associated with increased activity in the medial prefrontal cortex extending into the anterior cingulate cortex, insula, and temporoparietal junction (TPJ). Moreover, the dorsomedial prefrontal cortex, lateral orbitofrontal cortex, and TPJ tracked trial-by-trial variations in relative performance as a function of contribution structure, independent of self-relevance. Together, these findings suggest that self-allocation bias in performance-based decisions is unlikely to arise from distorted neural encodings of performance. Instead, self-interest may shape how contribution-structured performance information is used in social-allocation choices, providing a more precise account of how self-serving behavior emerges in cooperative contexts.

**Data availability statement:** All data and analysis codes related to this paper are available and archived in Open Science Framework at https://doi.org/10.17605/OSF.IO/AFDQS.

**Funding:** Y.M. was supported by the National Natural Science Foundation of China (projects 32430041, 32125019, https://www.nsfc.gov.cn/english/site_1/index.html), Brain Science and Brain-like Intelligence Technology-National Science and Technology Major Project (2022ZD0211000, https://en.most.gov.cn) and the Fundamental Research Funds for the Central Universities (2233300002, https://en.moe.gov.cn). C.W.K. was supported by the Emmy Noether Research Group grant (392443797, https://www.dfg.de/en) from the German Research Foundation (DFG). S.Z. was supported by a scholarship from the China Scholarship Council. For the publication fee we acknowledge financial support by Heidelberg University.The funders had no role in study design, data collection and analysis, decision to publish, or preparation of the manuscript.

**Competing interests:** I have read the journal's policy and the authors of this manuscript have the following competing interests: YM is a member of PLOS Biology's editorial board. The authors declare that they have no other competing interests.

**Abbreviations**: ACC, anterior cingulate cortex; amPFC, anterior medial prefrontal cortex; dmPFC, dorsomedial prefrontal cortex; fMRI, functional magnetic resonance imaging; FoV, field of view; FWE, family-wise error; GLMs, general linear models; HPD, Highest Posterior Density; IOFC, lateral orbitofrontal cortex; LOO, Leave-one-out; mPFC, medial prefrontal cortex; MTG, middle temporal gyrus; OforS, other-performance for self-allocation; SforO, self-performance for other-allocation; SVO, Social Value Orientation; TE, echo time; TI, inversion time; TPJ, temporoparietal junction; TR, repetition time.

## Introduction

Social cooperation is crucial for human interactions and navigating collective challenges. Individuals team up to locate hidden resources, complete complex projects, and solve collective problems. This cooperative behavior is often motivated by the promise of shared rewards upon successful achievement. Central to the sustainability of social cooperation is how individuals determine the distribution of these shared rewards. However, whether and how individuals allocate joint rewards based on individuals' varying performance and contribution remains largely unexplored. In this study, we aim to understand how individual performance, contribution, and social preferences shape such allocation decisions.

Two key processes are involved in such performance-based allocation decisions: assessing individual contributions from performance and allocating collective gains. The process of assessing each individual's contributions is conceptually linked to judgments of shared responsibility, where individuals evaluate each member's credit or blame for a group outcome based on their performance and the causal structure of the task [1–4]. While research on shared responsibility has delineated the psychological processes of attribution [4–6], less is known about how these assessments are translated into resource distributions. To investigate the allocation of collective gains, we adapted economic games, which have been widely used to quantify allocation decisions with numeric payoffs. Economic games such as the dictator game [7], ultimatum game [8,9], and third party punishment game [10], typically require participants to make monetary allocation choices between themselves and another person, measuring the degree to which individuals are willing to sacrifice personal earnings to increase or decrease those of others [11]. However, in these games, participants allocate gains that are independent of prior performance or contribution. Such windfall gains are rare in everyday life; as are the opportunities or necessities to share them with others. In most social interactions, individuals first cooperate, accrue benefits through successful cooperation, and subsequently determine the distribution of rewards based on prior performance and contribution. Similar to previous studies, which link efforts and rewards [12,13], the current study drew inspiration from social scenarios, where joint outcomes were designed to reflect participants' contributions within cooperative contexts. This approach allowed us to investigate the assessment of individual contributions based on their prior performance in collective tasks.

Steiner's theory on collective performance provides a valuable framework for categorizing collective tasks based on how individual performance contributes to the overall group performance and to reaching a common goal [14,15]. These collective tasks include additive and disjunctive task criteria. If the additive criterion applies, collective performance equates to the sum of individual performances (e.g., each person's speed in a torch relay or strength in a tug-of-war). In a "simple" version of this criterion, this summed performance directly translates into the joint outcome. More interesting versions of the "additive" criterion become relevant if a certain threshold must be reached (e.g., the runners in the torch relay must be faster than a given time). In the following, we will use the term "additive" only if the additive performance has to exceed a threshold and use the term "simple" otherwise. According to the

disjunctive criterion, collective performance mirrors the best individual performance (e.g., each person's knowledge in a pub quiz, [14]).

In our study, we designed a novel task incorporating varying levels of performance as well as simple, additive, and disjunctive task criteria. This structure, reflecting the taxonomy of collective task criteria, directly influenced individual contributions to joint outcomes. Specifically, we investigated how the nature of collective task criteria impacted social allocation decisions regarding shared rewards. To bridge the psychological processes of contribution assessment with the economic problem of reward division, we turned to game theory. Game-theoretic solution concepts like Shapley values [16,17] provide rational benchmarks for assessing contributions in collective tasks based on individual performance and can reflect the fairness and efficiency of allocation decisions. Thus, such game-theoretic concepts are suitable to understand scenarios in which participants collaborate to achieve a common goal. Shapley values can be calculated using a straightforward formula and specify each individual's added value in the group according to marginal contribution calculations [16]. To illustrate the concept, consider the following examples: In situations in which both individuals contribute equally to achieving the goal, a fair distribution is even. Conversely, if only one individual is solely responsible for accomplishing the goal, the rewards could be exclusively assigned to that person. In more intricate scenarios, Shapley values offer graded and quantitative benchmarks.

However, human behavior often deviates from such game-theoretic benchmarks [18]. A key driver of such deviations is the pervasive influence of self-serving biases [15,19,20]. This well-established tendency for individuals to process information in a self-advantageous manner manifests not only in attribution (e.g., taking credit for success and denying blame for failure) but also in behavior, such as exerting less effort for others' benefit [21]. We propose that this general self-serving tendency should also powerfully extend to the domain of resource distribution, leading to a systematic 'self-allocation bias'—where individuals overweight their own interests when dividing jointly earned rewards. This bias is expected to be modulated by self-relevance. Previous studies have demonstrated that people tend to perceive themselves and their future more positively than they perceive others [22–26]. Self-related biases also manifest in allocation decisions, with people typically favoring their future selves over others. For example, Molouki and Bartels found that individuals consistently allocated higher resources to their future selves compared to allocations made for others [27]. Building on this, our study investigated how self-relevance influences allocation decisions in performance-based contexts. Specifically, we compared "self-relevant" conditions, where performance and allocations directly involved the participants themselves, with "self-irrelevant" control conditions, where performance and allocations were related to other individuals within the same peer group. We hypothesized that participants would exhibit a self-related bias when allocating joint gains between the self and another individual, i.e., assign more rewards to themselves. Conversely, when allocating joint outcomes among two other individuals, we anticipated participants would treat others impartially and make allocations based on their performance. Additionally, we expected an interaction between self-relevance and collective task criteria, with the magnitude of a self-allocation bias varying across additive and disjunctive task conditions. Second, we sought to examine whether the expected self-related bias was related to the extent to which individuals would consider their own performance or contribution compared to that of others when making allocations. We hypothesized that participants relied more on prior performance of others in self-irrelevant situations than their own performance in self-relevant situations to make allocations. By systematically assessing the two aspects of allocation and performance that could contribute to a potential self-allocation bias, our study aimed to provide deeper insights into the mechanisms driving such a bias in performance-based allocation decisions.

Individual differences such as social preferences, have been shown to impact social decision-making [28–31], and provide an additional layer of insight into allocation behaviors. Social Value Orientation (SVO) is a crucial concept in understanding variations in social preferences, as it reflects an individual's concern for another person's material payoffs or well-being [32]. Based on SVO scores, individuals can be broadly categorized into individualistic persons, who prioritize self-interests and aim to predominantly maximize their own payoffs, and prosocial persons, who consider the payoffs of

others to a large degree. Previous studies have found connections between an individual's social preference, as indicated by the SVO scores, and their social decision-making regarding potential self-other allocations [33–35]. Therefore, we hypothesized that participants with lower SVO scores (indicated as more individualistic tendencies) would exhibit stronger self-related biases in allocation decisions, while participants with higher SVO scores (indicated as more prosocial tendencies) would make more balanced allocations.

Crucially, we investigated the neural correlates of such a self-allocation bias during performance-based social allocation. Making self-relevant allocation decisions requires the computation of self-relevant value and the consideration of the other person's claim. Midline regions of the brain, such as the medial prefrontal cortex (mPFC), the anterior cingulate cortex (ACC), and the posterior cingulate cortex, have been shown implicated in processing self-referential information and subjective value [36–43]. We therefore expected these regions to be engaged when allocating joint outcomes for the self as compared to allocating joint outcomes for others. The anterior insula, which reacts to social norm violation and inequity [44–46], might be involved in the trade-off between self-interest and fairness when allocating joint outcomes in a self-serving way. Furthermore, the need to consider the other individual's perspective in a cooperative setting led us to expect engagement of brain regions typically associated with evaluating others and integrating self-other information during social decision-making, such as the mPFC, bilateral temporoparietal junction (TPJ), superior temporal sulcus, and temporal pole [43,47–51]. The spatial organization within mPFC subserves distinct functional specialties [52]. The anterior medial prefrontal cortex (amPFC) activity reflects knowledge about the structure of the environment [53], which is expected to represent the different collective task structures in the present study, while the dorsomedial prefrontal cortex (dmPFC) is closely linked to social cognition [51,54]. Previous studies have shown that the dmPFC integrates the performance of self and other in cooperative tasks [55,56] and computes relative subjective values for self and other [57], we thus expected that the dmPFC would track each individual's performance and transform the performance into subjective values to allocate joint outcomes. Therefore, by implementing functional magnetic resonance imaging (fMRI), we aimed to examine neural activity associated with performance-based social allocation decisions. We employed a conservative, whole-brain analysis as our primary approach to test these hypotheses concerning these broad, functionally heterogeneous regions.

## Behavioral results

To characterize behavioral and neural processes associated with performance-based social allocation decisions, we conducted two complementary experiments: a behavioral study (Experiment 1) and an fMRI study (Experiment 2). Participants in both experiments engaged in two tasks: a performance task (i.e., a prime number task) followed by an allocation task (i.e., a modified dictator game; see Fig 1). To ensure a credible social context, four participants were scheduled simultaneously and tested in pairs, under the belief that their partner was one of the other participants and that task performance depended on a random subset of earlier trials in the performance task. A control analysis confirmed that the results remained robust in a sub-sample when excluding participants who were influenced by doubts about task manipulation (see Methods and Supporting information (SI) S6 and S7 Figs).

In both experiments, two conditions were set to examine the influence of self-relevance on allocation decisions. First, the condition "self-relevance" comprised two levels: In the self-relevant condition, participants were "participators," i.e., they made allocation decisions concerning joint outcomes for a self-relevant pair of players: themselves and another individual (self-performance for recipient-self). In the self-irrelevant condition, participants were "spectators," i.e., the allocation of joint outcomes was decided for a self-irrelevant pair of players: two other individuals (other-performance for recipient-other, Fig 1C). Crucially, participants, i.e., the "self," were always the persons who made the allocation decision. Second, the condition "collective task criteria" comprised three levels: Allocation decisions were contingent on prior performance according to three types of collective task criteria: simple, additive, and disjunctive. For conciseness, in the main text, we present the behavioral results based on a combined dataset across the samples of Experiments 1 and 2. Separate analyses lead to the same results and are presented in the SI; S1 and S2 Figs, S1–S4 Appendices). Given



# A Performance task

# B

**Self-relevant** **Self-irrelevant**

# C Allocation task

Fig 1. Experimental task design in Experiments 1 and 2. Participants completed two tasks sequentially: a performance task (prime number judgement) followed by an allocation task (modified dictator game). **(A)** In the performance task, participants judged whether a series of numbers were prime or not. Their responses were later used to represent task performance in the allocation task, with a random subset of correct responses selected for each trial. **(B)** Four participants completed the experiment simultaneously and were tested in pairs. From each participant's perspective, they were paired with one other participant (player Q) in the self-relevant condition, and the remaining two participants (players M and N) formed another pair in the self-irrelevant condition. In the self-relevant condition, participants made allocation decisions between themselves and player Q based on their respective performances. In the self-irrelevant condition, they allocated between players M and N based on those players' performances. **(C)** Example task frame for the three collective task criteria (in rows) under the self-relevant and self-irrelevant conditions (in columns). In the simple task, joint outcomes were determined by the total number of correct responses without a specific threshold. In the additive task, rewards were earned only if the sum of both players' correct responses exceeded the threshold. In the disjunctive task, rewards were earned if at least one player's correct responses exceeded the threshold. The experiment used a block design: each block began with an instruction specifying the condition (self-relevant vs. self-irrelevant) and the collective criterion ("no goal" for simple, "mutual reach" for additive, or "one reaches" for disjunctive). In each trial, participants were presented with information about the target goal (e.g., mutual correct responses ≥ 5), the performance of each player (e.g., player M had 2 correct responses, player N had correct 3 responses), the resulting joint outcome to allocate (e.g., win ¥10). They then adjusted a slider allocate between two players with a time limit (8 s in Experiment 1; 6 s in Experiment 2).

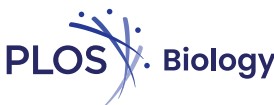

that total allocation and performance varied across trials, subsequent analyses were conducted using relative measures of allocation and performance to enable standardized comparisons. Thus, the relative allocation refers to the allocation to player 1 (i.e., participants themselves in self-relevant conditions) divided by the total allocation amount (player 1 plus player 2). Bayesian generalized non-linear multilevel models were then employed in all behavioral analyses to estimate the hierarchical effects of interested variables while appropriately accounting for the data distribution of these relative measures, see Methods for detailed descriptions. All models converged with R-hat = 1, suggesting that the samples sizes were sufficient for reliable inference.

**Identification of a self-related bias in allocation decisions.** To test our hypothesis concerning whether participants display a self-related bias on allocation decisions and whether collective task criteria impact the magnitudes of such a self-related bias, we employed a Bayesian multilevel model to predict participants' decisions of relative allocation by including fixed effects of self-relevance (self-relevant, self-irrelevant), collective task criteria (simple, additive, disjunctive), and their interaction, together with a random intercept of subjects (see Equation 1 in Methods). From posterior estimates, we observed a significant effect of self-relevance (Estimate = −0.19, SD = 0.03, 95% CI: [−0.24, −0.14], Table A in S1 Appendix). Specifically, post-hoc pairwise tests revealed that participants allocated significantly more rewards to the self in the self-relevant condition compared to the player 1 in self-irrelevant condition, with a median odds ratio of 1.21 (95% Highest Posterior Density (HPD) interval: [1.15, 1.28]) in the simple task condition, 1.25 (95% HPD: [1.19, 1.32]) in the additive task condition, and 1.23 (95% HPD: [1.17, 1.30]) in the disjunctive task condition, supporting our hypothesis of a self-related bias in allocation, Fig 2A, Table B in S1 Appendix. In contrast, we did neither observe clear differences nor in different task criteria and interactions between task criteria and self-relevance (all 95% CIs included zero). Post-hoc pairwise tests yielded odds ratios near zero and 95% HPD intervals overlapping 1, indicating no meaningful differences in relative allocation among the

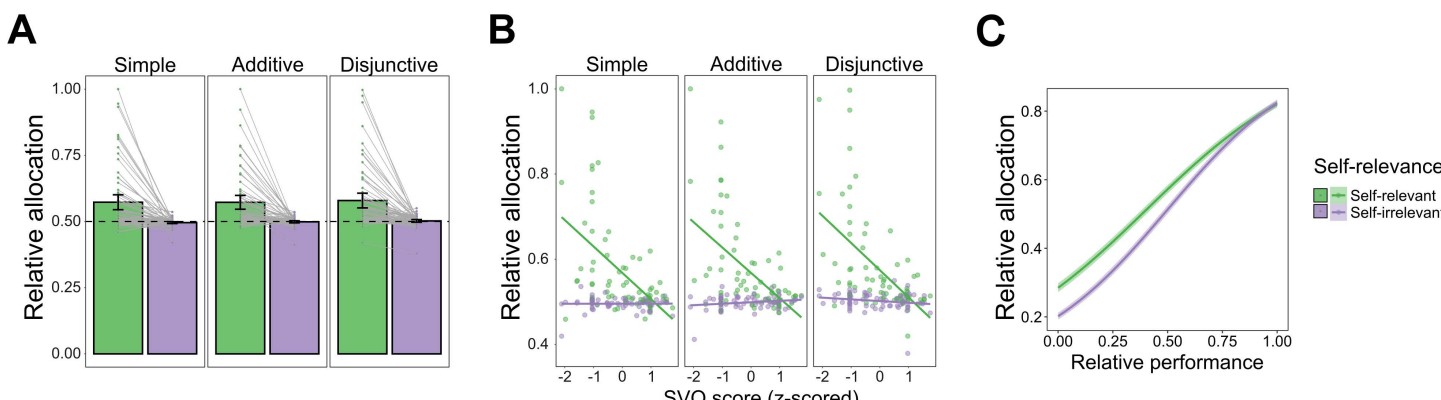

**Fig 2. Self-related bias in performance-based allocations and its relationship to Social Value Orientation (SVO). (A)** Participants allocated a larger proportion of rewards to the self-relevant player compared with the self-irrelevant player across all collective task criteria. Relative allocation was defined as the proportion allocated to player 1 (i.e., participant in the self-relevant condition and player M in the self-irrelevant condition) relative to the total allocation amount. Bars indicate group means; dots represent individual participants' average allocations; error bars represent standard errors The horizontal dashed line at 0.5 denotes the objective allocation expected from the symmetrically generated performance distributions. **(B)** SVO scores were negatively associated with relative allocation in all self-relevant conditions, but not in self-irrelevant conditions. Pairwise comparisons confirmed that the SVO slopes differed significantly between self-relevant and self-irrelevant conditions for all collective task criteria. **(C)** Relative performance more strongly predicted allocation decisions in the self-irrelevant than in the self-relevant condition. Shaded bands represent 95% credible intervals for the model-estimated trends. *Note: Panels (A) and (B) display each participant's mean allocation for visualization only; all statistical inferences were derived from hierarchical Bayesian multilevel models applied to the full trial-level data.* The data and code used to generate this figure are available at https://doi.org/10.17605/OSF.IO/AFDQS.

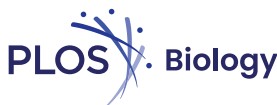

three task criteria or interaction effects. Additionally, we compared this model with a simpler model containing only the main effects with Leave-one-out (LOO) cross-validation and the comparison revealed that this more complex model provided a better fit to the data (difference in expected log predictive density(Δlpd) = −2.0, SE = 0.8). Overall, these results indicate that the allocation decisions were primarily influenced by self-relevance. This consistent difference between self-relevant and self-irrelevant condition provides clear empirical evidence for the self-allocation bias, which reflects a self-serving tendency to favor oneself in resource distribution.

Furthermore, it is worth noting that the parametrized performance presented to participants in the allocation task was intentionally designed to be symmetrically distributed for each pair of players. In theory, if participants were making allocation decisions objectively according to each individual's performance, the average of relative allocation across trials should converge around 0.5. Posterior estimates indicated that allocation ratios were significantly higher than 0.5 in all the self-relevant conditions (all 95% HPD intervals above 0.5), but not in the self-irrelevant conditions (all 95% HPD intervals included 0.5), Table C in S1 Appendix. These results further underscore that the self-allocation bias is solely driven by the biased allocations in self-relevant conditions but not the self-irrelevant conditions.

**The self-allocation bias and its correlation with Social Value Orientation (SVO).** Motivated by previous studies showing a modulation effect of SVO on social allocations [33,35,58], our second hypothesis aimed to test the influence of individual differences in SVO on allocation decisions. By including SVO as an additional fixed effect (see Equation 2 in Methods), we observed a significant interaction between SVO scores and self-relevance on relative allocation (Estimate = 0.16, SD = 0.03, 95% CI: [0.10, 0.21], Table A in S2 Appendix). Post-hoc pairwise tests revealed that the slopes of SVO in the self-relevant condition were consistently smaller than the slope in the self-irrelevant conditions across collective task criteria (Fig 2B, Table B in S2 Appendix). Specifically, posterior estimates of slopes indicated that SVO was negatively associated with relative allocation in the self-relevant condition (95% HPD intervals smaller than zero in all tack criteria). In contrast, no significant association was found between SVO and relative allocation in the self-irrelevant condition (all 95% HPD intervals included zero), Table C in S2 Appendix. These findings suggest the SVO selectively influences self-related allocation decisions, but not self-irrelevant allocations. As lower SVO scores reflect more individualistic social preferences and higher SVO scores reflect more prosocial preferences, our results suggest that individualistic participants showed a larger self-allocation bias of allocating more joint rewards to themselves.

**The self-allocation bias and its link to less reliance on performance.** To gain a deeper insight into the self-allocation bias, we examined our hypothesis on how this bias was reflected in performance. Building upon our earlier finding above that participants primarily considered and incorporated performance into allocation decisions in self-irrelevant conditions (as indicated by the comparison to the average relative allocation benchmark of 0.5), we anticipated a stronger link between performance and allocation in self-irrelevant conditions compared to self-relevant conditions. Thus, we employed another Bayesian multilevel model by including relative performance, self-relevance, task criteria, and two-way interactions as fixed effects (see Equation 3 in Methods) to analyze how individual performance affected participants' allocation decisions. Model comparison indicated that this model fitted best compared to a more complex model with a three-way interaction term and a simpler model without interaction terms (complex: Δlpd = −4.0, SE = 0.9, simple: Δlpd = −118.4, SE = 18.2). The slope in the model quantified as the weight of performance, indicates the extent to which participants based their allocation decisions on performance. Posterior estimates revealed that the slopes of relative performance in all conditions were significantly higher than 0 (all 95% HPD intervals above 0, Table B in S3 Appendix), which suggested that overall participants took performance into account. Post-hoc pairwise tests showed that the slope in the self-relevant condition was significantly lower than the slope in the self-irrelevant condition (Estimate = −0.57, 95% HPD: [−0.68, −0.46]), Fig 2C. Additionally, there was also a significant effect of task criteria that the slope increased from the simple task to additive task and to disjunctive task (all 95% HPD intervals below 0), Table C in S3 Appendix. These findings suggest that the self-allocation bias

might stem from performance-based allocation decisions, specifically by showing less reliance on performance when participants themselves are involved.

In our current design, self-relevance is tied to both who performs and who receives the allocations. Consequently, the self-allocation bias cannot be disentangled as arising from the source of performance or from the recipient of allocations, since both always refer to the self (self-performance for self-allocation, SforS) in the self-relevant condition and to another person (other-performance for other-allocation, OforO) in the self-irrelevant condition. To further extend our understanding of the self-allocation bias, we conducted an additional Experiment 3 (N = 62) to specifically examine the separate impacts of the "allocation recipient" and "performance source" that led to the outcomes to-be-allocated. In brief, we structured self-relevance into two distinct factors: performance source and allocation recipient, by including additional two self-relevance conditions: other-performance for self-allocation (OforS) and self-performance for other-allocation (SforO).

We built an initial full model with three within-subject factors and their interactions as fixed effects: performance source (self-relevant, self-irrelevant), allocation recipient (self-relevant, self-irrelevant), and collective task criteria (simple, additive, disjunctive), testing for the potential effects of self-relevance from performance source or allocation recipient. Posterior estimates revealed a significant effect of allocation recipient (Estimate = −0.14, SD = 0.03, 95% CI: [−0.19, −0.08]), while no other effects were found (all 95%CI included zero), Table A in S7 Appendix. The results indicated that participants' reward allocation only depended on whether the allocation target was the self or not. Additionally, model comparison revealed that a simpler model with a single fixed effect of allocation target showed better fit compared to the full model (Δlpd = −8.8, SE = 1.7), which further confirms that the effect of self-relevance originated from allocation target. Therefore, the findings indicate that participants care more about whom to allocate than based on whose performance. To be comparable with Experiments 1 and 2, we also plotted the average relative allocation for all conditions, S5A Fig. Post-hoc pairwise tests revealed that significant differences were found between two SforX (SforS and SforO) conditions, or between two OforX (OforS and OforO) conditions (all 95% HPD intervals above 1). However, no significant differences were found between XforS (SforS and OforS) conditions, or between two XforO (SforO and OforO) conditions (all 95% HPD intervals included 1), Table B in S7 Appendix. These results demonstrate a stronger self-allocation effect in allocation recipients than in performance source since the allocation decisions didn't differ wherever the performance was from the self or from the others, as long as the allocation was for the self or not. Additionally, by looking at the posterior estimates to compare relative allocations with 0.5, allocation ratios were significantly higher than 0.5 in three self-relavant (SforS, SforO, and OforS conditions (all 95% HPD intervals above 0.5, though lower 95% HPD = 0.501 in SforO condition), but not in the OforO condition (95% HPD intervals included 0.5), Table C in S7 Appendix. Taken together, these findings revealed that the bias favoring self-benefiting allocations was more strongly driven by preferences regarding allocation to oneself rather than evaluation of one's own performance. This experiment provides another angle into self-allocation bias, suggesting that the observed self-allocation bias stems more from a tendency to allocate more resources to oneself rather than from a biased evaluation of one's own performance.

**The effect of contribution categories on the self-allocation bias.** Based on different collective task criteria, each player's performance can be further transformed into their contribution to the joint outcome, which corresponds to the optimal solutions according to the Shapley value [16,17]. We identified three possible individual contribution structures to the joint outcome across the three collective tasks: *player 1 only*, *both*, and *player 2 only* (see Methods for details, Fig 3A). These structures separated whether joint rewards were received due to the performance of (i) player 1 (self or M) alone, (ii) both players, or (ii) player 2 (Q or N). Contribution influenced participants' allocation behavior, with a clear pattern emerging: participants allocated more to player 1 when player 1 was the primary contributor (i.e., in the "player 1 only" condition), and less to player 1 when player 2 was the primary contributor (i.e., in the "player 2 only" condition, S3 Fig). To directly test the effect of contribution on the self-allocation bias, we used a model with contribution as the fixed effect, and the self-allocation bias as the dependent variable, defined as the trial-wise difference in relative allocation between

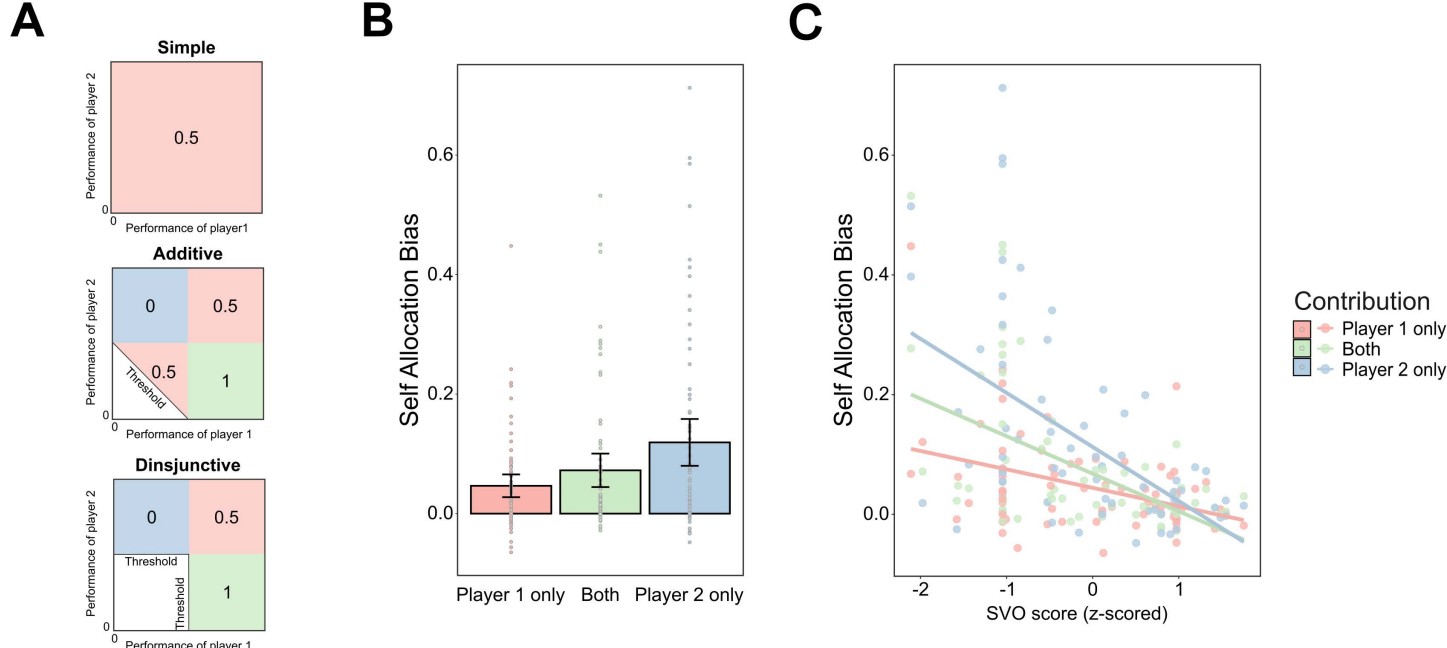

**Fig 3. Contribution categorization and its effect on self-allocation bias. (A)** The categorization of contributions with respect to Shapley values determines "optimal" reward allocation according to individual contribution based on collective task criteria. Rewards are distributed exclusively in the "colored" areas. The values shown within colored squares denote the allocation ratios assigned to player 1, as calculated by the Shapley value. Contribution structure is classified into three categories: a value of 1 indicates rewards are allocated entirely to player 1 ("player 1 only", yellow area); a value of 0.5 indicates rewards are shared equally between two players ("both", orange area); and a value of 0 indicated rewards are allocated entirely to player 2 ("player 2 only", blue area). **(B)** The self-allocation bias was larger in the "player 2 only" condition than in the "both" and the "player 1 only" conditions. The self-allocation bias was calculated as the relative allocation difference between self-relevant and self-irrelevant conditions. That is, the bias was highest when participants' performance did not contribute to the joint outcome. **(C)** In all contribution structures, SVO scores were negatively associated with the self-allocation bias showed that participants with lower SVO scores had a larger self-bias. The slopes decreased from "player 1 only" condition to the "both" condition, and then to the "player 2 only" condition. *Note: Panels (B) and (C) display the self-allocation bias averaged across trials within each participant for visualization only; all statistical inferences were derived from hierarchical Bayesian multilevel models applied to the full trial-level data.* The data and code for panels (B) and (C) are available at https://doi.org/10.17605/OSF.IO/AFDQS.

---

self-relevant and self-irrelevant conditions (see Equation 4 in Methods). Posterior estimates revealed that the self-allocation bias was presented across all contribution structures (all 95% HPD above zero, Table C in S4 Appendix). Crucially, we found a significant effect of contribution such that the self-allocation bias was strongest when only player 2 contributed to reaching the collective task threshold compared to situations where both players or only player 1 contributed (95% HPD intervals below zero for all pairwise comparisons, Fig 3B, Table D in S4 Appendix). This result suggests that the contribution structure modulates self-allocation bias, amplifying self-serving decisions when the partner is solely responsible for achieving the joint outcome.

Based on our earlier findings that individual differences in social preferences strongly correlated with the self-allocation bias, we further examined the relationship between SVO scores and the self-allocation bias with respect to contribution. Incorporating the SVO as an additional fixed effect (see Equation 5 in Methods). improved the model fit (Δlpd = −34.8, SE = 9.2). Posterior estimates indicated negative associations between SVO scores and self-allocation bias in all contribution structures (all 95% HPD intervals below zero), indicating that participants with lower SVO scores showed a larger self-allocation bias (Fig 3C, Table E in S4 Appendix). Notably, we observed an interaction effect showing that the slope decreased from "player 1 only" to "both", and then to "player 2 only" conditions (95% HPD intervals above zero for all pairwise comparisons, Table F in S4

Appendix). In summary, contribution had a substantial impact on allocation decisions: If the contribution of the partner alone was sufficient to reach the threshold, i.e., if the threshold was only reached through the partner's contribution, participants exhibited a stronger self-allocation bias, particularly those with more individualistic orientations.

## Neuroimaging results

Our behavioral results demonstrated a reliable self-allocation bias that was modulated by the contribution structure. To investigate the neural correlates underlying this bias and the role of performance and contribution, we analyzed fMRI data using three complementary general linear models (GLMs). GLM1 implemented a factorial design dissociating self-relevance and contribution structure while modeling relative performance as a parametric modulator to capture trial-by-trial variations in performance-related neural activity (detailed in Methods). GLM2 tested whether neural activity tracked trial-by-trial variation in allocation magnitude. GLM3 examined whether relative performance was encoded independently of task contexts. All reported clusters were whole-brain family-wise error (FWE) corrected for multiple comparisons at $p < 0.05$ with a cluster-defining threshold of $p < 0.001$. Full activation reports are given in S5 Appendix.

**Neural correlates of self-relevance and contribution structure (categorical effects).** We first examined categorical neural responses associated with allocation decisions by contrasting self-relevant versus self-irrelevant conditions (self-relevant > self-irrelevant in GLM1). This contrast revealed increased activity in regions commonly implicated in self-related processing and social cognitive functions [37–39,59,60], including the superior frontal gyrus/mPFC extending into the ACC (peak voxel in MNI coordinates $x$, $y$, $z$: 18, 54, 24), right insula lobe/anterior insula (34, 16, −10), and right angular gyrus within the TPJ complex (46, −56, 32; Fig 4, Table A in S5 Appendix). No significant activations were observed for the reverse contrast (self-relevant < self-irrelevant).

In contrast, no significant brain activations were found for the main effect of contribution or for the interaction between self-relevance and contribution at the categorical level. These results indicate that self-relevance robustly modulates the overall decision state during allocation, whereas contribution structure does not produce categorical differences at decision onset.

**Contribution-dependent encoding of relative performance (parametric modulation).** We next examined whether relative performance was parametrically encoded during allocation decisions and whether this encoding depended on contribution structure. In GLM1, relative performance was entered as a parametric modulator separately for each contribution condition ("Player 1 only", "Both", "Player 2 only").

Onset: Self-relevant > Self-irrelevant

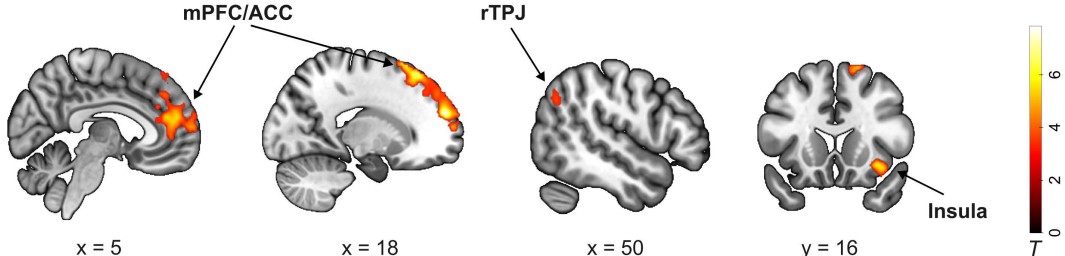

**Fig 4. Neural activity for the categorical effect of self-relevance.** Contrasting onsets of self-relevant vs. self-irrelevant conditions showed activations in the superior frontal gyrus/medial prefrontal cortex extending into the anterior cingulate cortex, right insula lobe/anterior insula, and right angular gyrus within the temporoparietal junction (TPJ) complex. No regions were found for the contrast of self-irrelevant > self-relevant. The group-level data for this figure are available at https://doi.org/10.17605/OSF.IO/AFDQS.

The results showed that the contribution structure influenced how relative performance was tracked on a trial-by-trial basis (i.e., contribution-dependent modulation of relative performance encoding) in superior frontal gyrus/dmPFC [12,28,52], bilateral middle orbital gyrus/lateral orbitofrontal cortex (lOFC, left: −42, 50, −6; right: 40, 48, −12), bilateral angular gyrus/TPJ (left: −56, −64, 30; right: 44, −64, 42), and right middle temporal gyrus (MTG: 64, −38, −4; Fig 5A, Table A in S5 Appendix).

To illustrate these effects, beta coefficients corresponding to relative performance were extracted separately for each contribution condition (Fig 5B). Across all regions, beta estimates were significantly lower in the "Player 1 only" condition compared with the other two conditions (*ps* < 0.05, S6 Appendix). Notably, beta estimates were negative in the "Player 1 only" condition, near zero in the "Both" condition, and positive in the "Player 2 only" condition, indicating a graded modulation across contribution structures.

Since the parametric modulator encodes both players' performance using relative rather than absolute performance of one player, this pattern does not reflect player identity. Instead, the sign of the coefficients within these regions potentially reflects tracking of the performance of the non-essential contributor, whose performance is not causally necessary for group success but is informative for resolving ambiguity in reward allocation. These findings indicate that relative performance is encoded in a contribution-structured but self-independent manner.

**Neural activity does not track allocation magnitude (GLM2).** To test whether neural activity tracked trial-by-trial variation in allocation outcomes, we conducted an alternative GLM (GLM2, Methods) in which self-relevance was modeled as an onset regressor and relative allocation was entered as a parametric modulator. The contrast for self-relevant versus self-irrelevant onsets again revealed brain activations similar to those observed with GLM1 (Table B in S5 Appendix). However, no significant brain activity was associated with the parametric modulation of relative allocation. These results

Parametric modulation: the effect of contribution levels

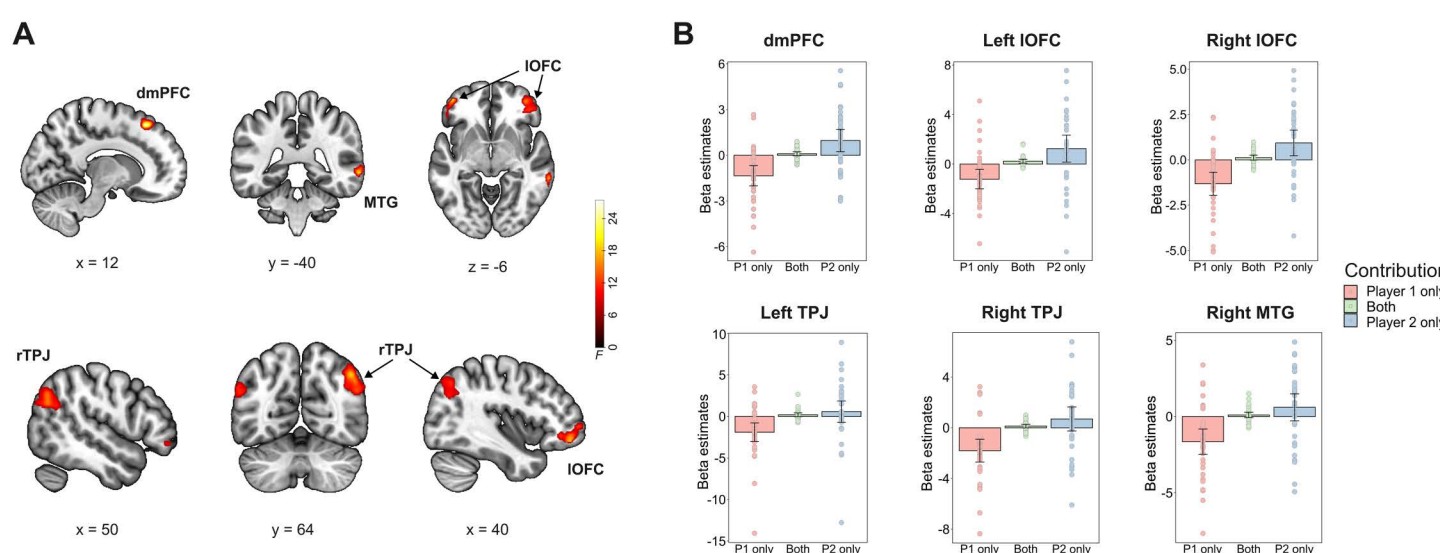

**Fig 5. Neural activity for the contribution-dependent encoding of trial-by-trial variations in relative performance. (A)** Activity in the superior frontal gyrus/dorsomedial prefrontal cortex, bilateral middle orbital gyrus/lateral orbitofrontal cortex, bilateral angular gyrus/TPJ, and right middle temporal gyrus reflected the effect of contribution structure on the trial-by-trial variations in performance. **(B)** To illustrate specific comparisons in (A), beta estimates of each contribution structure within the functional ROIs were plotted. The group-level data for panel (A) and the individual values and code for panel (B) are available at https://doi.org/10.17605/OSF.IO/AFDQS.

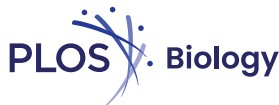

suggest that neural responses during allocation primarily reflect the cognitive context of self-relevant decision-making rather than the magnitude of the allocation outcome itself.

**No context-independent encoding of relative performance (GLM3).** Finally, we examined whether relative performance was encoded independently of self-relevance and contribution structure using a simplified GLM (GLM3) that included relative performance as a single parametric modulator without condition regressors. This analysis did not reveal any significant performance-related activations. Together with the GLM1 results, this finding indicates that performance-related neural signals emerge only when relative performance is embedded within the structure of contribution, rather than being encoded as a context-independent quantity.

Together, the neuroimaging results revealed dissociated neural encoding of self-relevance and relative performance during allocation decisions. Self-relevance modulates the overall decision state, whereas relative performance is encoded in a contribution-dependent manner but not specific to self-relevance.

## Discussion

The equitable distribution of collective rewards following group cooperation is essential for sustaining long-term collaboration. While previous research has extensively examined how people make self-other economic allocations in simplified paradigms [28,35,61] and responsibility is evaluated in group contexts [2–4], less is known about how objective performance information and causal contribution are encodes and transformed into allocation behavior. The present study bridged this gap by investigating the cognitive and neural mechanisms underlying performance-based allocations in collective tasks. Across experiments, we showed a robust self-allocation bias particularly when their performance did not directly contribute to the joint outcomes. This bias intensified among more individualistic individuals. Our neural findings revealed dissociated encoding self-relevance and performance during allocation decisions. Self-relevance modulates the engagement of mPFC, ACC, insula, and TPJ regions during the overall decision state, whereas dmPFC, lOFC, MTG and TPJ regions are involved in tracking trial-by-trial relative performance in a contribution-dependent but self-neutral manner.

At the behavioral level, participants prioritized self-interest when allocating rewards between themselves and others, whereas allocations were more impartial in self-irrelevant contexts. This pattern extends well-established research demonstrating that people typically prioritize their own benefits over others by exerting more efforts [21] and enlarging self-interest in social decisions [11,32]. Our experiments show that this tendency remains within a novel social context of performance-based decision-making. The self-allocation bias was strongly associated with social preferences: more individualistic participants exhibited stronger self-serving allocation bias, whereas prosocial participants allocated more equally. Notably, these individual differences predicted behavior only when the self was directly involved, indicating that social preferences shape biased allocation decisions specifically when the self is involved. These findings are consistent with prior research showing that individuals are not purely self-interested and that heterogeneity in social preferences, such as SVO, plays an important role in social decision-making [32,62,63].

Accurately accessing both own and others' performance and ability is crucial during group decision-making [64,65]. In the present study, the introduction of prior performance represents an important but previously overlooked factor, which allows us to examine how objective performance metrics and task contributions are integrated into allocation decisions. Our findings revealed that the self-allocation bias appeared not only as a baseline preference for self-benefitting allocation distribution but also as a reduced reliance on own actual performance. In other words, individuals prioritized personal advantage over objective performance information to distribute resources in a self-serving manner. Therefore, the self-allocation bias examined in the present study can be understood as a specific and behaviorally measurable manifestation of the broader self-serving bias, extending it from attributional judgements to concrete decisions of resource distribution. Evidence from Experiment 3 further supports this interpretation. By disentangling two potential mechanisms where

self-allocation bias may come from, results in Experiment 3 provided evidence that the observed bias stemmed primarily from a preference for oneself as the recipient of rewards rather than from an inflated evaluation of own performance.

A methodological advance of the present study is the use of the Shapley value to operationalize contribution-based fairness in cooperative tasks [16,17]. By providing an objective and causally grounded measure of individual contribution, this approach allows performance-based norms to be specified independently of subjective impressions or interpersonal preferences. Behaviorally, participants generally respected Shapley-derived norms by rewarding high contributors. However, deviations from these norms emerged when self-interest was at stake: individuals allocated themselves disproportionate rewards even when their causal contribution to the joint outcome was minimal. This pattern reveals a tension between explicit fairness norms and self-serving motivations, such as self-enhancement [19,66] or entitlement [67,68].

The neural results provided insights into these possibilities. When contrasting self-relevant with self-irrelevant allocation decisions, we observed increased activity in the mPFC (extending into the ACC), insula, and TPJ. These regions have been implicated in self-related processing, social decision-making, and evaluating or comparing self to others [37,39,40,61,69–73]. Engagement of this network suggests that self-relevant allocation decisions are characterized by heightened motivational and social-cognitive demands, including the evaluation of outcomes for oneself in relation to others. In particular, prior work has linked mPFC activity to self-referential evaluation and motivation [44], TPJ activity to perspective-taking and comparison between self and others [63,71–75], and insula and ACC activity to affective salience and norm-related conflict [46,76–79]. Together, these findings indicate that allocating rewards to oneself versus to others recruits regions supporting the arbitration between self-interest and social considerations. Notably, additional analyses (GLM2) showed no parametric modulation of these regions by allocation magnitude, suggesting that their engagement primarily reflects the self-relevant decision context rather than graded differences in the specific allocation outcome.

A complementary pattern emerged when examining how relative performance was represented neurally. Performance-related activity in medial and lateral prefrontal regions, as well as temporoparietal cortex, was detectable only when contribution structure was modeled (GLM1), but not in a simplified model that ignored contribution structures (GLM3). This dissociation indicates that these regions do not merely encode performance in a generic manner; instead, they encode performance information in relation to the causal structure of the task. Specifically, trial-by-trial variation in relative performance was tracked in a contribution-dependent fashion in dmPFC, lateral OFC, TPJ, and MTG. Across all regions, neural responses exhibited a consistent graded pattern: beta coefficients were negative when only player 1 determined the outcome, near zero when both players contributed, and positive when only player 2 determined the outcome. Crucially, this graded pattern may reflect sensitivity to the performance of the player whose contribution was *non-essential* for the group outcome. When only one player causally determined success, activity in these regions covaried with the performance of the other player, the one whose performance was irrelevant for producing the outcome but remained informative for deciding how rewards should be allocated. This finding suggests that, even in situations where contribution is asymmetrical, allocation decisions rely on a comparative encoding of both players' performance rather than a narrow focus on the causal contributor alone. Such encoding is consistent with prior evidence implicating dmPFC in tracking self and other performance during social decisions [18,55,56,80], lateral OFC in representing task structure and contextual features relevant for valuation [81–83], and TPJ in integrating information about others' actions and contributions during cooperative interactions [84–88].

Taken together, these neural results may suggest that performance information is encoded in a structured but largely self-neutral manner, shaped by contribution structure rather than by self-relevance per se. This stands in contrast to the behavioral findings, where allocation decisions were biased toward the self in self-relevant contexts. One implication of this dissociation is that self-serving behavior in performance-based allocations may not require distorted or self-favoring encoding of performance at the neural level. Instead, self-bias may emerge at a later stage, when contribution-structured performance information contributed to concrete allocation choices and weighted by self-relevance. The evidence that the self-allocation bias comes from the self-relevance in allocation target rather than performance source from Experiment

3 provides potential behavioral support for this idea. Although the temporal resolution of fMRI precludes direct separation of representational and decisional stages, the present findings constrain the locus of self-bias by showing that it is not expressed as a pervasive self-centered encoding of performance. Alternatively, the explicit presentation of objective performance may reduce the opportunity for biased performance evaluation and thus contributed to the absence of a self-bias effect on performance representation. A central question for future work to address is how self-relevance modulates the transformation of performance information into allocation decisions, for instance by reshaping subjective performance evaluation, or reweighting related motives such as fairness and contribution. Answering this question requires combining refined task designs that increase variability and incorporate subjective evaluations, enabling computational models that disentangle distinct motivational drivers of trial-by-trial allocation behavior, with methods using higher spatio-temporal resolution, such as intracranial recordings.

Additionally, future studies could expand on our findings by exploring the self-allocation bias in different populations. The large individual variability in self-allocation bias and the targeted neural correlates have potential implications for clinical fields, particularly in understanding and addressing maladaptive social behaviors. Clinical research can leverage our individual differences framework of self-allocation biases to study disorders marked by pathological self-biases (e.g., narcissism, depression). Studies that further explore differences between ingroup and outgroup dynamics or varying the interpersonal closeness of recipients [89–91] could provide insights into how social relationships shape decisions in performance-based allocations. A limitation of the present study is the use of simulated stimuli rather than real interaction during the task. Although our task design eliminated the need for direct interaction which is common in social fMRI tasks due to practical constraints and the need for experimental control, such approach still reduces the ecological validity of social interaction [92]. Ensuring participants' engagement and belief in the simulated social context is crucial for capturing neural and behavioral signatures of social cognition processes [93]. Addressing these considerations would help gain a more comprehensive understanding of the underlying mechanisms driving self-allocation bias under various social and cooperative contexts.

In conclusion, our study provides additional evidence linking economic game studies to real-world allocation scenarios by investigating allocation decisions under the specific context of cooperation in collective tasks. By connecting variations of group performance to later reward allocations, we offer a nuanced perspective on how such decisions unfold in a more realistic setting. Our behavioral results identify a self-allocation bias, characterized by increased self-allocation and reduced reliance on own performance when decisions involve self-relevant joint outcomes. The magnitude of the self-allocation bias is modulated by individual social preferences and the level of contribution within the group. Neural evidence highlights the involvement of self-referential processing and social decision-making networks, mainly in the midline and temporoparietal regions during the allocation decision process. Together, the present study refines current understanding of self-serving behavior in cooperative contexts. Self-allocation bias appears to reflect how the encoded performance information may be used for social allocation when self-interest is involved, which does not necessarily require biased neural encoding of performance. This highlights the importance of separating performance representation from allocation decision processes in models of social decision-making for future work.

## Methods

### Participants

Two main experiments (Experiment 1: lab; Experiment 2: fMRI) and a third additional experiment (Experiment 3: lab) were conducted in the present study. All participants were university students recruited through online advertisement. All participants had normal or corrected-to-normal vision and reported no history of neurological or psychiatric diagnoses. Participants were not currently taking medication and reported no drug or alcohol abuse. In Experiment 1, 36 male participants (mean age = 22.26 years, SD = 1.94, range 18–26, age information of two participants missing due to a recording

error) were recruited. In Experiment 2, one participant was excluded from all analyses due to an incorrect stimuli list presented in the allocation task, resulting in a final sample of 35 participants (18 females, 17 males, mean age = 21.41 years, SD = 2.42, range 18–28, age information of three participants missing). One participant was excluded from further fMRI data analysis due to excessive (beyond 3 mm) head motion during scanning. In Experiment 3, data of three participants were excluded (one participant reported psychiatric disorder history after task completion) or missing (two participants), resulting in 62 male participants (mean age = 22.54 years, SD = 2.40, range 18–29, age information of six participants missing) as the final sample.

## Ethics statement

The study was conducted in accordance with the Declaration of Helsinki and approved by the ethics committee of the State Key Laboratory of Cognitive Neuroscience and Learning at Beijing Normal University (No. ICBIR_A_0107_002). All participants gave written informed consent before the experiment and received monetary compensation for their participation.

## Experimental procedures

All experiments consisted of two main tasks: the performance task and the allocation task (Fig 1A). Both tasks were presented on computer screens using MATLAB psychtoolbox-3 (www.psychtoolbox.org). To establish a credible social context, four participants were scheduled to arrive simultaneously and were tested in pairs. Each participant was led to believe that their task partner was one of the other individuals with whom they had arrived, and that performances were based on a random subset of trials from their earlier task performance, independent of their overall performance. Additionally, part of each participant's compensation was also determined by a random selection of trials in the allocation task, and this procedure was explicitly explained to them during the practice session.

**Performance task.** The performance task required participants to decide if numbers were prime numbers or not (the concept of prime numbers was clearly explained to participants). All participants received the same lists of numbers for three collective task criteria, but in different orders. Each list contained 40 numbers ranging from 1 to 1,000: 20 prime numbers, 10 even numbers, and 10 odd non-prime numbers. For each number, participants had 3 s to make their choice. If no response was given within 3 s, the trial was considered incorrect. Participants were led to believe that their number of correct responses would determine their performance in the upcoming allocation task.

**Allocation task.** After completing the performance task, participants undertook the allocation task, a modified version of the dictator game. In the task, participants made monetary allocations between two pairs of individuals based on their prior performance in the performance task. As shown in Fig 1B and 1C, the allocation task included two within-subject factors: self-relevance and collective task criteria. First, two conditions were involved regarding self-relevance: self-relevant and self-irrelevant. In the self-relevant condition, participants acted as "participators" and made allocations for a pair of players including themselves and another individual (alias Q). In the self-irrelevant condition, participants acted as "spectators" and made allocations for a pair of other two individuals (aliases M and N). Participants were informed that Q, M, and N were other participants in the experiment, either from the same or another testing day. Responses were coded as the amounts of money (¥) participants allocated to *player 1* (self or M) and *player 2* (Q or N). Second, we designed collective task conditions that followed three distinct criteria: (i) "Simple" criterion: In this condition, each pair of players received shared rewards based on the individual number of correct responses from each player. (ii) "Additive" criterion: Here, the goal of getting joint rewards was for the collective performance, i.e., the total number of correct responses from both players, to exceed a given threshold that varied across trials. (iii) "Disjunctive" criterion: Joint rewards were given if at least one player's performance (the higher number of correct responses) reached the threshold. To make the task more realistic, we also included loss trials in addition to those where joint rewards were shared, but these trials were not included in the main analyses.

Participants were led to believe that in each trial, both players' performances were determined by the accuracy in 20 out of 40 prime judgments and that the 20 prime judgments were randomly selected from the performance task. Therefore, performance in each trial could reasonably vary. The actual performance shown in the allocation task was parametrically manipulated in advance. We generated a list of trials and pseudo-randomized their order for each participant. The list was the same for the self-relevant and self-irrelevant conditions and symmetrically distributed between player 1 and player 2 within each pair to ensure equal contribution (i.e., the average performance was the same for each player).

In both Experiment 1 (384 trials in four sessions) and Experiment 2 (216 trials in six sessions), mini blocks appeared alternately within sessions for every experimental condition. These blocks began with instructions describing the specific collective task criterion and self-relevance condition. Within each block, trials were presented in a pseudo-randomized order across participants. Each trial displayed the threshold (left blank in the simple task), the two players' performance, and the joint outcome in one single stage. In Experiment 2, participants had a maximum of 6 s to make the allocation decision in each trial, Fig 1C. In Experiment 1, the time limit was 8 s. If a participant did not make the decision within time limit, the trial was considered a miss trial (miss trial ratio: $0.12 \pm 0.0004\%$ for Experiment 1, $0.96 \pm 0.02\%$ for Experiment 2).

Before the formal allocation task, participants performed a short practice session of 12 trials. After finishing the allocation task, participants completed the SVO slider task [94] and answered a few post-check questions including their believability of the task manipulation, strategy, and emotions during the task. For each item of the SVO slider task, participants had to choose their most preferred option from nine options of payoffs for themselves and another person. We calculated SVO scores from 6 primary items.

**Experimental design in additional Experiment 3.** This experiment introduced two additional conditions to examine self-relevance depending on performance source and allocation recipient separately. Specifically, in one additional condition, allocations were made between self and Q according to the prior performance of M and N. We define this condition as the Other-for-Self (OforS) condition, i.e., relying on others' (M and N's) performance to make allocation for self and Q. Conversely, in another additional condition, allocations were made between M and N according to the prior performance of self and Q, which we define as the Self-for-Other (SforO) condition, i.e., relying on own and Q's performance to make allocations for others (M and N). The original self-relevant and self-irrelevant conditions were thus named as the Self-for-Self (SforS) and Other-for-Other (OforO) conditions in this experiment (S4 Fig). In such cases, the dependence on performance and allocation can be separated. That is, both the performance was determined by either the self and another person (i.e., SforX) or by two other people (i.e., OforX), and the two potential recipients could include the self (i.e., XforS) or they could be two other people (i.e., XforO).

## Behavioral data analysis

We used the relative allocation ratio of player 1 to account for variability in performance and allocation across trials. Specifically, we divided the performance or allocation of player 1 (participants themselves in the self-relevant condition, or M in the self-irrelevant condition) by the total performance or allocation of both players. This approach standardizes and quantifies the measurement of both performance and allocation [95]. Consequently, the relative performance or relative allocation of two players is always summed up to 1. As the relative allocation has zero-one-inflated beta distribution, we implemented Bayesian generalized non-linear multilevel models to test our hypotheses by using the "brms" package in R Studio with the family of zero_one_inflated_beta(). This method stabilizes parameter estimation and improves effective power in small samples or unbalanced data, which shows significant advantages over traditional mixed models that are highly sensitive to sample size [96–98]. Bayesian estimation was performed using 8 chains of 3,000 iterations each and 100 warmups. Model convergence was assessed via R-hat of parameters, which indicates whether effective samples sizes were sufficient for reliable inference. Posterior estimates and post-hoc tests were conducted using the function emmeans() and emtrends() to examine the significance of the regression coefficients and differences between conditions



using estimated marginal means. Model comparisons were assessed using LOO cross-validation by examining the differences in log predictive density and the associated standard error.

We combined the two experiments for behavioral data analysis since they shared the same experimental design. Separate results for each experiment are reported in the SI (S1 and S2 Figs, S1–S4 Appendices). The experimental design includes two within-subject factors: self-relevance (self-relevant, self-irrelevant) and collective task criteria (simple, additive, disjunctive). In the interest of transparency, we note that the study was not preregistered. However, the research hypotheses were developed prior to data collection but were not formally preregistered.

To test our hypothesis regarding a self-related bias in allocation, we build a first model including fixed effects of self-relevance, and collective task criteria, along with their potential interactions on the relative allocation ratio of player 1.

$$relative\ allocation \sim relative\ performance * self\_relevance + relative\ performance * task\ criteria + \left(1\ |\ participant\ ID\right)$$
(1)

A simpler model without the interaction term was tested and compared with the first model to assess model fit.

To test the hypothesis regarding the relationship between a self-related bias and SVO scores, we added SVO score as an additional fixed effect into the first model. SVO scores were standardized before being included in the analysis.

$$relative\ allocation \sim self\_relevance * task\ criteria * SVO\ score + \left(1\ |\ participant\ ID\right)$$
(2)

To test the hypothesis regarding self-related bias in performance, models including relative performance and the two factors as fixed effects were fitted to evaluate the extent to which participants' allocation decisions were influenced by individual performance. Model comparisons indicated that the model including two-way interaction terms provided the best fit.

$$relative\ allocation \sim relative\ performance * self\_relevance + relative\ performance * task\ criteria + \left(1\ |\ participant\ ID\right)$$
(3)

After calculating Shapley values using relative ratio (see Fig 3A), the transformation of Shapley values into relative contributions categorized reward trials into three contribution structures: (a) "Player 1 only": reaching the collective task goal depended solely on the performance of player 1 (participants themselves or M). The Shapley value suggested allocating the full joint rewards to player 1, i.e., the allocation ratio is 1 for player 1 and 0 for player 2. (b) "Both": reaching the threshold required the performance of both players. The Shapley value prescribes an even allocation, i.e., 0.5 for each). (c) "Player 2 only": achieving the threshold relied exclusively on the performance of player 2 (Q or N), i.e., 1 for player 2 and 0 for player 1. Specifically, for the simple task, we considered all trials to be in the "both" condition because there is no actual threshold in this task. Since no effect of collective task criteria was found on the self-related bias, all task criteria were combined to spilt trials into contribution strcutures. Table A in S4 Appendix summarized the distribution of trial numbers according to these contribution categorizations. To further examine the effect of contribution on the self-allocation bias, we fitted a model with contribution as the fixed effect and self-allocation bias as the dependent variable.

$$self\_allocation\ bias \sim contribution + \left(1\ |\ participant\ ID\right)$$
(4)

The self-allocation bias was quantified as the difference between self-relevant and irrelevant conditions. We also tested the correlations between SVO scores and the self- allocation bias for each contribution structure by further including the SVO score as a fixed effect.

$$self\_allocation\ bias \sim contribution * SVO\ score + \left(1\ |\ participant\ ID\right)$$
(5)



Post-experiment check revealed that a high percentage of participants across all experiments (86.5% on average) reported that their decisions were not influenced by whether the other players in the task were real or fictitious. We performed a rigorous control analysis to exclude participants who expressed doubts about the reality of the partners and reported that this belief influenced their decisions (Experiment1: 5 excluded; Experiment 2: 7 excluded; additional Experiment 3: 6 excluded). All our key behavioral results remained statistically unchanged in these purified samples (S6 and S7 Figs).

## Brain imaging acquisition

Whole-brain images were acquired using a Siemens 3.0 Tesla TRIO MRI scanner at the Brain Imaging Center of the National Key Laboratory of Cognitive Neuroscience and Learning at Beijing Normal University. Functional brain images were collected using a T2-weighted gradient-recalled echo planar imaging sequence (slices = 37, repetition time [TR] = 2,000 ms, echo time [TE] = 30 ms, flip angle = 90 degrees, field of view [FoV] = 224 × 224 mm, slice thickness = 3.5 mm, voxel size = 3.5 × 3.5 × 3.5 mm$^3$). High-resolution structural images were obtained with a T1-weighted sequence (slices = 144, TR = 2,530 ms, TE = 3.39 ms, inversion time [TI] = 1,100 ms, flip angle = 7 degree, FoV = 256 × 256 mm, slice thickness = 1.33 mm, voxel size: 1.3 × 1.0 × 1.3 mm$^3$). To correct image distortions, a fieldmap (slices = 37, TR = 500 ms, TE1 = 3.38 ms, TE2 = 5.84 ms, slice thickness = 3.5 mm, voxel size = 3.5 × 3.5 × 3.5 mm$^3$) was acquired prior to the experimental run.

## fMRI data analysis

**Preprocessing.** Imaging data analyses were performed using SPM12 (Statistical Parametric Mapping 12; Welcome Department of Imaging Neuroscience; www.fil.ion.ucl.ac.uk/spm). Imaging preprocessing steps included geometric distortion correction with field map images, slice timing correction, realignment for head motion correction, co-registration, spatial normalization into Montreal Neurological Institute (MNI) space (voxel size = 2 × 2 × 2 mm$^3$), and smoothing with a 6 mm full-width at half-maximum Gaussian kernel.

**General linear model.** Three GLMs were set up to investigate the neural correlates of performance-based allocation decisions. The first GLM (GLM1) examined the influence of contribution and self-relevance across six conditions, specified by the following onset regressors: self-relevant: player 1 only, self-relevant: both, self-relevant: player 2 only, self-irrelevant: player 1 only, self-irrelevant: both, self-irrelevant: player 2 only. Loss trials were included as a regressor of no interest. We concatenated all sessions into one session to eliminate onsets that were empty for individual sessions. We examined how neural signals encoded performance by including the relative performance as a parametric modulator of onset regressors in GLM1. Furthermore, to assess neural markers associated with the magnitude of relative allocation, we specified a second GLM (GLM2) that included the onset regressors for self-relevance, with relative allocation entered as a parametric modulator for each onset regressor. A third GLM (GLM3) was constructed with a single onset regressor encompassing all trials and relative performance was included as a parametric modulator to examine brain activity related to a basic effect of performance independent of other factors.

The decision stages in the allocation task were entered as events with a duration of 6 s. Regressors were convolved with a canonical hemodynamic response function. Six motion-correction parameters estimated from the realignment procedure were entered as covariates of no interest. We generated multiple contrast images using the spm_make_contrasts function at the first level. Then, using these images, one-sample *t*-tests for factors with two levels and two-sample t-tests for factors with three levels at the second level were conducted to detect regions showing main effects and interactions of the factors. Since the mPFC is a relative large and functionally heterogeneous regions and the TPJ is not strictly anatomically defined [99–102], we employed robust and conservative whole-brain analyses as the primary method to test our priori hypotheses regarding these regions, while remaining open to exploratory findings in other brain areas. Through this approach, we aimed to provide a comprehensive understanding of the neural mechanisms underlying performance-based

social allocation decisions. For all whole-brain analyses, we set the statistical threshold of all reported clusters at a FWE corrected threshold of $p < 0.05$ at the cluster-defining threshold of $p < 0.001$.

## Supporting information

**S1 Fig. Self-related bias in performance-based allocations and its relationship with SVO score in experiments 1 and 2, respectively.** Panels **(A–C)** present results from Experiment 1, and panels **(D–F)** present results from Experiment 2. All statistical analyses were conducted using hierarchical Bayesian multilevel models, and the results remained robust across analyses. The data and code used to generate this figure are available at https://doi.org/10.17605/OSF.IO/AFDQS.
(TIFF)

**S2 Fig. Self-allocation bias and its association with SVO scores across contribution structures in experiments 1 and 2 separately.** Panels **(A, B)** show results from Experiment 1, and panels **(C, D)** show results from Experiment 2. All statistical tests were conducted using hierarchical Bayesian multilevel models and the results remained robust. The data and code used to generate this figure are available at https://doi.org/10.17605/OSF.IO/AFDQS.
(TIFF)

**S3 Fig. Self-relevant and self-irrelevant average relative allocation in three contribution structures separately in the combined sample.** The data and code used to generate this figure are available at https://doi.org/10.17605/OSF.IO/AFDQS.
(TIFF)

**S4 Fig. Conceptualization and examples of additional conditions in Experiment 3. (A)** Conceptual diagram of the two additional conditions introduced in Experiment 3. The SforS (self-performance for self-allocation) and OforO (other-performance for other-allocation) conditions correspond to the self-relevant and self-irrelevant conditions in Experiments 1 and 2, respectively, where self-relevance in both performance source and allocation recipient is consistent. The two additional conditions: SforO (self-performance for other-allocation) and OforS (other-performance for self-allocation), introduce crossed combinations of self-relevance between performance source and allocation recipient. **(B)** Example task frames for the OforS and SforO conditions in Experiment 3.
(TIFF)

**S5 Fig. Relative allocations, their associations with SVO scores, and self-allocation bias across contribution structures in Experiment 3. (A)** Relative allocations across all conditions. **(B)** Associations between SVO scores and relative allocations. **(C)** Slopes of relative performance predicting relative allocation. **(D)** Self-allocation bias across three contribution structures. **(E)** Associations between self-allocation bias and SVO scores. For illustration, data from the four conditions are plotted separately. In the actual analysis, these four conditions were modeled using two factors (allocation target and performance source) within hierarchical Bayesian multilevel models. The effect of allocation target was consistent with the self-relevance effect observed in Experiments 1 and 2. The data and code used to generate this figure are available at https://doi.org/10.17605/OSF.IO/AFDQS.
(TIFF)

**S6 Fig. Relative allocations, their associations with SVO scores, and self-allocation bias across contribution structures in Experiment 1 and 2 after excluding participants who expressed doubts about the reality of the partners and reported that this belief influenced their decisions. (A)** Relative allocations across all conditions. **(B)** Associations between SVO scores and relative allocations. **(C)** Slopes of relative performance predicting relative allocation. **(D)** Self-allocation bias across three contribution structures. **(E)** Associations between self-allocation bias and SVO scores. All

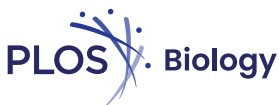

statistical tests were conducted using hierarchical Bayesian multilevel models and the results remained robust after exclusion. The data and code used to generate this figure are available at https://doi.org/10.17605/OSF.IO/AFDQS. (TIFF)

**S7 Fig. Relative allocations, their associations with SVO scores, and self-allocation bias across contribution structures in Experiment 3 after excluding participants who expressed doubts about the reality of the partners and reported that this belief influenced their decisions. (A)** Relative allocations across all conditions. **(B)** Associations between SVO scores and relative allocations. **(C)** Slopes of relative performance predicting relative allocation. **(D)** Self-allocation bias across three contribution structures. **(E)** Associations between self-allocation bias and SVO scores. All statistical tests were conducted using hierarchical Bayesian multilevel models and the results remained robust after exclusion. The data and code used to generate this figure are available at https://doi.org/10.17605/OSF.IO/AFDQS. (TIFF)

**S1 Appendix. All statistical analyses related to the model (Equation 1) testing the effect of self-relevance and collective task criteria on relative allocation (Equation 1) in the combined sample, or Experiment 1 and 2, respectively.**
(DOCX)

**S2 Appendix. All statistical analyses related to the model (Equation 2) testing the effect of self-relevance and collective task criteria on the relationship between SVO and relative allocation in the combined sample, or Experiment 1 and 2, respectively.**
(DOCX)

**S3 Appendix. All statistical analyses related to the model (Equation 3) testing the effect of self-relevance and collective task criteria on the relationship between relative performance and relative allocation in the combined sample, or Experiment 1 and 2, respectively.**
(DOCX)

**S4 Appendix. All statistical analyses related to the models (Equations 4 and 5) testing the effect of contribution and SVO score on self-allocation bias in the combined sample, or Experiment 1 and 2, respectively.**
(DOCX)

**S5 Appendix. GLM results.**
(DOCX)

**S6 Appendix. Statistical tests for the difference in beta estimates from ROI of contribution structures.**
(DOCX)

**S7 Appendix. All statistical tests for the additional Experiment 3.**
(DOCX)

## Acknowledgments

We thank Gabriela Rosenblau, Lisa Doppelhofer, and Benjamin Kuper-Smith for their helpful comments on earlier versions of the manuscript.

## Author contributions

**Conceptualization:** Yina Ma, Christoph W. Korn.

**Data curation:** Xue Yong.

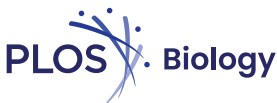

**Formal analysis:** Sihui Zhang.

**Funding acquisition:** Yina Ma, Christoph W. Korn.

**Methodology:** Sihui Zhang, Christoph W. Korn.

**Supervision:** Yina Ma, Christoph W. Korn.

**Visualization:** Sihui Zhang.

**Writing – original draft:** Sihui Zhang, Christoph W. Korn.

**Writing – review & editing:** Sihui Zhang, Yina Ma, Christoph W. Korn.

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
