## [Editor Report · Decision Letter 0]

17 Jun 2025

Dear Dr Zhang,

Thank you for submitting your manuscript entitled "The self-allocation bias – behavioral and neural characterization of a newly identified bias for individual performance and contribution in cooperative tasks" for consideration as a Research Article by PLOS Biology.

Your manuscript has now been evaluated by the PLOS Biology editorial staff and I am writing to let you know that we would like to send your submission out for external peer review. Please note, however, that we were unable to receive advice from an Academic Editor and have some concerns regarding the novelty of the findings. We will seek comments on this aspect from the reviewers and will discuss it with an Academic Editor after review.

Once your full submission is complete, your paper will undergo a series of checks in preparation for peer review. After your manuscript has passed the checks it will be sent out for review. To provide the metadata for your submission, please Login to Editorial Manager (https://www.editorialmanager.com/pbiology) within two working days, i.e. by Jun 19 2025 11:59PM.

Kind regards,

Christian

Christian Schnell, PhD

Senior Editor

PLOS Biology

cschnell@plos.org

---

## [Decision Letter · Decision Letter 1]

20 Aug 2025

Dear Dr Zhang,

Thank you for your patience while your manuscript "The self-allocation bias – behavioral and neural characterization of a newly identified bias for individual performance and contribution in cooperative tasks" was peer-reviewed at PLOS Biology. It has now been evaluated by the PLOS Biology editors, an Academic Editor with relevant expertise, and by several independent reviewers.

In light of the reviews, which you will find at the end of this email, we would like to invite you to revise the work to thoroughly address the reviewers' reports.

As you will see below, the reviewers are overall supportive of this study. Reviewer 1 and Reviewer 3 have a few concerns about the statistical analyses and the integration with the existing literature. They also suggest a few additional analyses. Reviewer 2 raises the small sample size and the potential lack of pre-registration as a major concerns. Having discussed these concerns with the Academic Editor, we think that you should focus on fully addressing the concerns from Reviewer 1 and Reviewer 3. We think that the replication within your study provides sufficient support for the reliability of the statistical results, but please address Reviewer 2's remaining concerns.

Given the extent of revision needed, we cannot make a decision about publication until we have seen the revised manuscript and your response to the reviewers' comments. Your revised manuscript is likely to be sent for further evaluation by all or a subset of the reviewers.

**IMPORTANT - SUBMITTING YOUR REVISION**

*Re-submission Checklist*

*Published Peer Review*

*PLOS Data Policy*

*Blot and Gel Data Policy*

Sincerely,

Christian

Christian Schnell, PhD,

Senior Editor

PLOS Biology

cschnell@plos.org

REVIEWS:

Reviewer #1: In this study, Zhang and colleagues investigated how human participants distribute resources (money) between two individuals—either themselves and another person, or two other people—based on each individual's performance in a separate task. They found an allocation bias whereby participants tended to allocate more money to themselves than to others. The authors then examined the neural mechanisms underlying this biased allocation process.

The study is interesting and potentially informative. However, I have serious concerns regarding the study's design, the statistical analyses (both behavioral and fMRI), and certain aspects of the conclusions drawn. I believe the authors should be given an opportunity to resubmit, as my comments are all addressable.

Below are my comments, which I hope the authors will find constructive and helpful.

Major comments:

1- The study is conceptually related to shared responsibility judgement, where people working in groups decide who to blame/credit for joint outcomes. This is very close to what this study investigates. However, never in the introduction or discussion is this link mentioned. In particular, some key references of the shared responsibility literature are missing. For example,

Xiang et al. Cognition 2023 https://www.sciencedirect.com/science/article/abs/pii/S0010027723002433

Gerstenberg and Lagnado 2010. https://psycnet.apa.org/record/2010-04148-013

Mahmoodi et al. 2024. https://www.sciencedirect.com/science/article/pii/S0896627324006494

Gerstenberg et al. 2023. https://www.sciencedirect.com/science/article/abs/pii/S0010027723001336

The introduction talks at length about experiments using games (game theory). I understand the link, the authors use one to study joint responsibility, but I do not see a clear conceptual link between this study and game theory.

2- The authors conducted linear regressions to examine the effect of relative performance on allocation across conditions (line 259 onward). They report using t-tests as their statistical test and then compared the resulting beta weights between conditions. This suggests that they did not use a mixed-effects linear regression. If this is the case, I strongly encourage the authors to reanalyze these data using a mixed-effects model, including condition as a fixed effect and testing the interaction between condition and relative performance. This approach would allow them to assess differences between conditions directly within a single model, while properly accounting for within-subject variability.

I do not expect this change to substantially alter the results, but it is important for ensuring that the findings generalize beyond the current sample. Simple t-tests only test for differences within the sample at hand, whereas mixed-effects modeling appropriately generalizes the inference to the broader population.

3- Similarly, a major issue in the data analysis is the use of t-tests and ANOVAs instead of linear regression models. For example, in the behavioral data analysis (e.g., line 200 onward), the authors compared performance weights to zero and conducted ANOVAs to compare weights across conditions. However, the authors could—and in my view should—perform their analyses using a single linear model, with allocation as the dependent variable and relative performance and condition(s) as independent variables, including all relevant interaction terms. This would allow the authors to examine the effects of all key factors simultaneously and to test for interactions directly.

This approach is important because it can also be applied to the neuroimaging analysis. Instead of conducting GLMs based solely on factorial designs, the authors could use parametric designs incorporating allocation, condition, and their interaction as regressors. This would make it possible to identify the unique contribution of each factor and any differences between them. In contrast, the current analytical approach introduces unnecessary confusion and potential confounds. For instance, the authors report greater activity in the mPFC, TPJ, ACC, and insula in the self-relevant compared to the self-irrelevant condition. However, the behavioral results show that allocations were higher in the self-relevant condition. Therefore, this analysis does not clarify whether the reported neural differences reflect the parametric effect of allocation or simply self-relevance.

Relatedly, I am also concerned about the way the neural GLMs were implemented. In one GLM (line 337 onward), the authors included relative performance as a parametric modulator (PM) at the individual level, and allocation bias at the group-level GLM. This approach is confusing and raises questions about its rationale. If the goal is to examine the effect of relative performance (RP) on brain activity, the GLM should include RP as a PM without modulating the group level by allocation bias. Why should the neural representation of RP depend on allocation bias? This implies an assumption that the way people represent relative performance is conditional on their allocation bias, but the authors do not provide a clear justification for this. Why not first run a simpler GLM with only RP as the parametric modulator?

If the simpler GLM does not yield significant results, and the authors' approach does, this should be clearly reported and discussed. Such a result does not compromise the study—it strengthens it by demonstrating that the more complex model is justified.

4-The self-allocation bias reported here is related to self-serving bias. Please make a link to it.

5- In Experiment 3, the authors report that the allocation bias is driven more by a tendency to allocate additional resources to oneself than by a tendency to overestimate one's own performance. First, it would be helpful to report the full results of this experiment. Did participants also show any evidence of overestimating their performance? Additionally, in the other experiments, participants were shown their actual performance—was this also the case in Experiment 3? If so, given that there was no ambiguity about performance, was it even possible to detect any effect of an inflated self-performance estimate?

6- "Thus, the activities of both mPFC and TPJ suggest an integration of self-referential processing and mentalizing networks for self-other allocation decisions." (Line 438). Can the authors clarify how their task is related to mentalising? Just because they found activation in TPJ/mPFC does not mean that their task requires mentalising. If the current task has an element of mentalising in it, the authors should mention that and explain why. Linking a process to mentalising just because it recruits the brain regions which light up during mentalising is reverse inference I am afraid.

Minor comments:

-Line 81: "task criteria impacted on social allocation…". There is an extra "on", it should be task criteria impacted social allocation…

-Line 100: "found that individuals allocated consistently higher resources to their future selves compared to allocations made for others." This sentence is convoluted. Does consistency refer to resources or allocation. If the latter, it should be individuals consistently allocated…, otherwise it should be clarified.

-Line 157 "reflects the structure knowledge of the environment". Was it meant to be "reflects knowledge of the structure of the environment"? or "reflects the structural knowledge of the environment"? please clarify.

-It will save reviewers quite a bit of time if the figures are placed in the text where they are relevant followed by their captions. Putting the text, the figures and the captions each in a different place does not help anyone.

-In addition, the quality of the figures is extremely low. Figure 1 is illegible. Please replace them.

-Line 260: "we examined in our third hypothesis". It should be "we examined our third.."

-Line 276: "So far, our design does not allow us to dissociate the self-relevance regarding performance source or allocation recipients since it consistently pertained to whose performance and allocations were considered." This sentence is incomprehensible. Please rewrite.

-Line 349 "Second, we tested whether activity varied according to individual differences in the self-allocation bias in the interaction between self-relevance and contribution levels. The result showed that the behavioral self-allocation bias was linked to the posterior..". I am not sure what it is that the authors tested here. Did you test the effect of the interaction or the effect of allocation bias? Please clarify.

Line 437: and comparing self with others others. There seems to be an extra others.

Reviewer #2: Korn et al. used a novel effort-based reward allocation task to study how self-allocation bias affects reward allocation in cooperative tasks and investigated the neural mechanisms underlying this process. The task itself is novel, and the behavioral results are interesting and compelling. The neural analyses use conservation statistical thresholds, which is great. However, there are major issues that make the manuscript unpublishable in its correct form in a high impact journal like PLOS Biology. First, and more importantly, the authors are making claims about human social behavior (and their neural processes) using a new, previously untested task design with very small sample sizes, and these effects were not preregistered (at least that was not stated in the paper). This is not acceptable scientific practice for social neuroscience in 2025. The neural effects also appear to be very exploratory though the introduction is written as if only specific hypotheses were being tested. An appropriate middle ground to allay concerns around reproducibility and exploratory analyses is to address the comments below and perform an independent behavioral replication in a much larger sample (see comment 1-2).

Major comments

1) I understand that the authors collected an initial behavioral sample and then re-ran the task for fMRI data collection. Since the second behavioral data set was a presumed replication, did the authors preregister these effects? If so, please state, and if not, please explain. Similarly, can the authors state whether any of the neural hypotheses were preregistered?

2) The sample sizes in this study are very small, which is especially concerning with a new social paradigm. Further, the practice of collecting a very small behavioral sample in study 1, and also in study 2, and then combining these samples in the main text seems misleading. To go forward with publishing the behavioral and neural results of a new economic task, the authors should show a pre-registered replication in a much larger sample of least N = 200. This would add a level of confidence in the reliability of the task and the authors claims.

3) Were the partners in the allocation task synthetic agents? If so, did participants think these were real partners? Did the authors check for believability of this manipulation?

4) I appreciate that the authors are trying to convey the task frames to readers to explain the task, but the figure itself fails to explain what the actual task was. What were the questions/prompts that people were attempting to answer in the scanner? Also, not sure if this is just an English translation issue, but to state "You corrected 3" is very confusing. Were the participants correcting errors as part of the task? If the authors instead mean to communicate how many answers were correct on a performance task, please include the actual language that was relayed to participants.

5) The neural results of this paper appear highly exploratory. Although the authors explain hypothesized involvement of self-referential/mentalizing brain regions, such as ACC, mPFC, dmPFC, and TPJ, the authors do not motivate the involvement of other regions known to contribute to social decision-making, such as ventral striatum and hippocampus, although these regions are described in the results. If the authors hypotheses were only related to regions described in the introduction, why did the authors conduct a whole-brain analysis rather than ROI-based analyses? I think the main issue is that the introduction is written as if the authors are tested very specific hypotheses, but the results are written as if the analyses conducted were exploratory. For instance, looking for individual differences in self-allocation bias in the hippocampus and ventral striatum seems very post-hoc and exploratory since neither of these subcortical regions, which of course have their own specializing functions, are theoretically motivated. So, either the introduction needs to state that there were some hypothesized effects, but the actually conducted analyses were exploratory, or the authors should state in the results sections which results and analyses were directly related to the main study hypotheses, and which were exploratory/serendipitous findings.

6) If I understand correctly, the parametric modulation analyses for the self-relevant conditions (player 1) effects are associated with reduced functional engagement of TPJ and dMPFC regions, as shown in Fig. 5. Is reduced functional activation of these regions the correct interpretation? In other words, are these results directionally consistent with the authors hypotheses? I could be wrong as the interpretation may just depend on the direction of the slope for the parametric modulation effect, so please clarify how the negative beta coefficients should be correctly interpreted.

Reviewer #3: The paper presents results of an fMRI experiment on social preferences. In the study, the participants perform a simple number sorting task and then make a series of monetary allocations. In one condition ("self-relevant"), these allocations are made between themselves and another person; in another condition ("self-irrelevant"), the allocation is made between two other people. The participant can use the results of the number sorting task to make the decision; there are 3 additional conditions: in the "simple" condition, the total reward is based on their individual performance; in the "additive" condition, it is based on some threshold their joint performance must reach; in the "disjunctive" condition, only one person had to reach the threshold.

The study finds that people tend to favor their own outcomes (as described in a large body of literature) but also tend to overlook their own poor task outcomes during allocations if that benefits them; this effect is stronger in people with low social value orientation. The paper then finds various neural correlates of certain decision variables (self-relevant vs self-irrelevant allocations, variations in performance, individual differences in self-allocation bias).

Strong qualities of the paper:

(1) The main idea of combining the social allocation with Shapley value-style contribution is novel and interesting.

(2) The results are clearly presented and the overall flow is easy to follow.

(3) The neuroimaging methods are clearly described and overall at the standard of the field.

Potential issues:

(1) As far as I could understand (it is not clearly described in the methods), the task uses deception (there are no real money allocations that are made) and the conditions are set up to fit the design. This is a common and significant issue in fMRI work in social neuroscience, as it is impossible to guarantee that participants base their decisions on actual preferences and not on experimenter demand effects. I do not see how this could be mitigated; I see this is a significant flaw that needs to be at least discussed, but of course, it's up to the editor to decide whether this flaw is critical for the publication in the journal.

(2) The paper mentions some theoretical constructs, such as social preference functions and Shapley values, but lacks a proper theoretical foundation. What are the hypotheses that are being tested based on? What are the decisions in each trial based on? There are many computational/utility models of similar decisions and it would greatly benefit the paper to set up even a simple model that would estimate the values of various choices on each trial.

(3) The paper mentions individual variability results in the neuroimaging section, which are presented as activation maps. Given that the paper explores the individual relationships between SVO and choices in Figures 2 and 3, it would make sense to also show some scatterplots relating these neural activations to these variables on the individual level.

(4) The activations labeled TPJ seem to be quite dorsal for TPJ and could potentially be IPS/IPL instead? These often co-activate with the lateral OFC. It might make sense to double check the regions and borders here.

(5) The figure panels could use a bit more labeling beyond the captions, as it can be difficult to determine which brain map corresponds to which contrast.

(6) The main neuroimaging results seem to only report positive activations. Were there no negative ones (it seems so from the supplementary table)? This is a bit unexpected and maybe deserves some mention/discussion.

---

## [Decision Letter · Decision Letter 2]

13 Jan 2026

Dear Dr Zhang,

Thank you for your patience while we considered your revised manuscript "The self-allocation bias – behavioral and neural characterization of a newly identified bias for individual performance and contribution in cooperative tasks" for consideration as a Research Article at PLOS Biology. Your revised study has now been evaluated by the PLOS Biology editors, the Academic Editor and two of the original reviewers .

In light of the reviews, which you will find at the end of this email, we are pleased to offer you the opportunity to address the remaining points from the reviewers in a revision that we anticipate should not take you very long. Please make sure to address the reviewer comments very carefully, in particular the concerns about the theoretical foundations for the decisions and the interpretation of your results in the discussion. Both reviewers shared these concerns and it will be important for future readers to have these concerns fully addressed in the revised manuscript. We will then assess your revised manuscript and your response to the reviewers' comments with our Academic Editor aiming to avoid further rounds of peer-review, although we might need to consult with the reviewers, depending on the nature of the revisions.

**IMPORTANT - SUBMITTING YOUR REVISION**

*Resubmission Checklist*

*Published Peer Review*

*PLOS Data Policy*

*Blot and Gel Data Policy*

Sincerely,

Christian

Christian Schnell, PhD

Senior Editor

PLOS Biology

cschnell@plos.org

REVIEWS:

Reviewer #1: Thank you for addressing my comments. It appears that the GLMs using task variables as parametric modulators do not yield meaningful results. What the fMRI data instead show is that certain brain regions are more active when participants allocate money to themselves compared with allocating money to others. I think the authors should clarify in the Discussion what these results do and do not imply, as well as the limitations of such a broad finding. For example, the observed self-relevance effect could be driven by expected reward in the self-relevant condition rather than self-related processing per se. As the results are currently presented, it is not straightforward—at least for me—to understand their precise interpretation.

Reviewer #3: I appreciate the effort the authors put in their revision. Many aspects of the methods and results were clarified, and my comments were mostly addressed. I noticed that other reviewers were somewhat critical, and I believe that the lack of a proper theoretical foundation for the decisions is an issue that could be important for a high-impact journal such as PLOS Biology. As the authors noted, this could not be addressed with the current design, so I can't recommend acceptance without hesitation, but overall, this is an interesting and potentially impactful paper.

---

## [Editor Report · Decision Letter 3]

6 Feb 2026

Dear Dr Zhang,

Thank you for your patience while we considered your revised manuscript "The self-allocation bias: behavioral and neural correlates of performance and contribution in cooperative tasks" for publication as a Research Article at PLOS Biology. This revised version of your manuscript has been evaluated by the PLOS Biology editors and the Academic Editor.

Based on our Academic Editor's assessment of your revision, we are likely to accept this manuscript for publication, provided you satisfactorily address the following data and other policy-related requests:

* We would like to suggest a different title to improve its accessibility for our broad audience. Would any of these two suggestions work for you?

** Prefrontal and temporoparietal circuits support self‑serving biases in performance‑based self allocation in cooperative tasks

OR

** Self-allocation bias in performance-based cooperative decisions is driven by self-interest rather than distorted performance encoding

* Please add the links to the funding agencies in the Financial Disclosure statement in the manuscript details.

* Please add a declaration in the competing interest section that Yina Ma is a member of PLOS Biology's editorial board.

* Please include information in the Methods section whether the study has been conducted according to the principles expressed in the Declaration of Helsinki.

* DATA POLICY:

Regardless of the method selected, please ensure that you provide the individual numerical values that underlie the summary data displayed in the following figure panels as they are essential for readers to assess your analysis and to reproduce it: 2A, 3B, 5B, S1AD, S2AC, S3, S5AD, S6AD and S7AD.

* CODE POLICY

Per journal policy, if you have generated any custom code during the course of this investigation, please make it available without restrictions. Please ensure that the code is sufficiently well documented and reusable, and that your Data Statement in the Editorial Manager submission system accurately describes where your code can be found. More information on our Code Policy, what and how to share can be found here: https://journals.plos.org/plosbiology/s/code-availability

* Please move the methodological details from the supplementary information to the main manuscript file. We have no word count limit and want to make it as easy as possible for readers to find the information they need.

We expect to receive your revised manuscript within two weeks.

*Published Peer Review History*

*Press*

Sincerely,

Christian

Christian Schnell, PhD

Senior Editor

cschnell@plos.org

PLOS Biology

---

## [Editor Report · Decision Letter 4]

23 Feb 2026

Dear Dr Zhang,

Thank you for the submission of your revised Research Article "Self-allocation bias in performance-based cooperative decisions is driven by self-interest rather than distorted performance encoding" for publication in PLOS Biology. On behalf of my colleagues and the Academic Editor, Matthew Rushworth, I am pleased to say that we can in principle accept your manuscript for publication, provided you address any remaining formatting and reporting issues. These will be detailed in an email you should receive within 2-3 business days from our colleagues in the journal operations team; no action is required from you until then. Please note that we will not be able to formally accept your manuscript and schedule it for publication until you have completed any requested changes.

PRESS

We frequently collaborate with press offices. If your institution or institutions have a press office, please notify them about your upcoming paper at this point, to enable them to help maximize its impact. If the press office is planning to promote your findings, we would be grateful if they could coordinate with biologypress@plos.org. If you have previously opted in to the early version process, we ask that you notify us immediately of any press plans so that we may opt out on your behalf.

Sincerely,

Christian

Christian Schnell, PhD

Senior Editor

PLOS Biology

cschnell@plos.org